# Abundant Genetic Diversity Harbored by Traditional Naked Barley Varieties on Tibetan Plateau: Implications in Their Effective Conservation and Utilization

**DOI:** 10.3390/biology13121018

**Published:** 2024-12-05

**Authors:** NiMa QuZhen, Lhundrup Namgyal, Dawa Dondrup, Ying Wang, Zhi Wang, Xing-Xing Cai, Bao-Rong Lu, La Qiong

**Affiliations:** 1Key Laboratory of Biodiversity and Environment on the Qinghai-Tibetan Plateau, Ministry of Education, School of Ecology and Environment, Tibet University, Lhasa 850000, China; m13659594010_1@163.com; 2Yani Observation and Research Station for Wetland Ecosystem of the Tibet (Xizang) Autonomous Region, Tibet University, Lhasa 850000, China; 3State Key Laboratory of Hulless Barley and Yak Germplasm Resources and Genetic Improvement, Research Institute of Agriculture, Tibet Academy of Agriculture and Animal Husbandry Sciences, Lhasa 850000, China; lhundrupnamgyal@163.com (L.N.); dwdunzhu@126.com (D.D.); 4Department of Ecology and Evolutionary Biology, Fudan University, Ministry of Education Key Laboratory for Biodiversity and Ecological Engineering, Shanghai 200438, China; wang_y@fudan.edu.cn (Y.W.); 18110700017@fudan.edu.cn (Z.W.); xxcai@fudan.edu.cn (X.-X.C.)

**Keywords:** cluster analysis, gene pool, genetic variation, germplasm preservation, hulless barley, molecular fingerprint, population structure, Tibet

## Abstract

The genetic diversity of naked barley (*Hordeum vulgare* var. *nudum*) varieties from Tibet, including traditional (T), modern (M), and germplasm-resources-bank (G) types, was assessed by analyzing SSR molecular fingerprints. The results show significant genetic diversity, particularly in the T varieties that contain a higher number of private (unique) alleles. Principal coordinates and STRUCTURE analyses reveal considerable deviation of M varieties from T/G varieties. Evidence of seed mixture in M varieties suggests the use of mixed seeds in contemporary cultivation practices. Cluster analysis further indicates that M varieties have a narrower genetic background, likely due to the limited number of T or G varieties that were used in naked barley breeding. Relationships between increases in genetic diversity and sample sizes highlight the importance of effective sampling strategies for field collections of naked barley varieties.

## 1. Introduction

World food security relies essentially on the sufficient production of food crops, of which genetic resources (also known as germplasm) play critical roles in the sustained genetic improvement of these crops [1,2,3]. Therefore, effective conservation and utilization of crop genetic resources will guarantee sustainable crop production and global food security [4,5,6]. Naked barley, also referred to as hulless barley, (*Hordeum vulgare* L. var. *nudum* Hook. f.) is an important food crop that is widely distributed in the world [7]. Naked barley is the same species, but a unique type of cultivated barley (*Hordeum vulgare* L.) [7]. The reason why this crop is referred to as naked barley is simply that the inner and outer glumes on spikes can easily be separated from caryopsis at the ripening stage, so that naked barley’s grains (commonly known as seeds) are naked or hulless [8], like the grains of common wheat. Naked barley is very important for people living in marginal and mountainous agroecosystems at relatively high elevations, such as on the North African Plateau, the Andes Plateau in South America, in Caucasus Mountain areas in Russia, and on the Qinghai-Tibet Plateau in China [9,10]. As a unique and valuable crop, naked barley therefore makes up an important component of global food security, particularly for people living the marginal areas.

In China, naked barley has been cultivated for more than 3000 years, although this crop has been cultivated mainly in mountainous areas with relatively harsh environmental conditions and high elevations, such as in Tibet Autonomous Region, Qinghai, Gansu, Sichuan, Yunnan, and Guizhou provinces [11]. Importantly, naked barley not only serves as the main human food and livestock feed, but is also used as the raw materials for manufacturing beers, local medicines, and various health-care products [12,13,14]. As the only staple food crop that can ripe, or mature, and produce seeds normally at high altitudes, naked barley can be cultivated on the alpine Qinghai–Tibet Plateau with an elevation at >4500 m above sea level. Therefore, naked barley also possesses great economic value with extremely important roles for people living in those areas [15]. Therefore, the traditional naked barley varieties (Figure 1, left) have been used in these agroecosystems for thousands of years, and modern varieties (Figure 1, right) have successively been produced to replace the traditional varieties with unwanted performances.

In addition, during the long-term domestication and evolution under harsh natural and artificial selection, naked barley has accumulated abundant genetic variation to adapt to different environmental conditions to respond to the rapid changes [7,11,14,16]. Such changeable environmental conditions, include different altitudes (1000~5000), soil types, ultraviolet radiations, etc., allowing naked barley to have developed a wide range of characteristics to adapt to the extreme conditions on the Tibet Plateau [10,11,17,18]. Apart from the adaptation to these harsh environmental conditions, naked barley has also developed the important agronomic trait for the relatively short growth period (early mature), which makes the naked barley varieties suitable to grow in barren and cold areas with relatively inadequate accumulative temperatures [7,19]. Consequently, this crop has established many ecological types with rich phenotypic variation [20,21] and genetic diversity [10,11,18,22,23] through adaptive evolution. Therefore, it is urgent to determine such genetic diversity categorized in different naked barley gene pools harbored by the traditional, modern, and germplasm-resources-bank stored germplasm or genetic resources.

However, limited studies have been carried out to address the genetic diversity of naked barley varieties, even though this crop is so unique and valuable, due to the above-mentioned characteristics. Therefore, it is very important to investigate the genetic diversity of the naked barley varieties both as a staple food crop and as valuable genetic resources in different gene pools. A better understanding of genetic diversity harbored by the naked barley varieties will facilitate the effective utilization and efficient conservation of the naked barley varieties by providing appropriate sampling strategies not only in Tibet but also in other regions of the world. Particularly, under future unpredictable scenarios caused by global climate change [16,24,25], the efficient conservation, evaluation, and utilization of naked barley genetic resources in a timely manner becomes even more important. This is because genetic diversity in naked barley accumulated by long-term domestication and human uses may be rapidly reduced and even lost under the changing environment and strong disturbances by human activities in particular [26,27].

To determine genetic diversity within and among naked barley varieties categorized in different gene pools, we studied 20 randomly collected naked barley varieties, representing the traditional and modern gene pools, respectively, from the Tibetan Plateau. In addition, we also studied 10 conserved naked barley varieties, representing the gene pool of the germplasm-resources-bank in Tibet. We applied the high put-through simple sequence repeat (SSR) molecular fingerprinting technology to characterize the genetic diversity of these naked barley varieties. The primary objectives of this study were to address the following questions: (i) Is genetic diversity harbored within the naked barley germplasm evenly distributed in the three representative gene pools of the Tibetan varieties? (ii) What are the genetic structures and relationships of the three representative gene pools of the Tibetan naked barley varieties? (iii) What is the relationship between increases in genetic diversity and sample size within a variety of the three representative Tibetan naked barley gene pools? Answers to these questions will provide insights into the effective conservation and utilization of naked barley germplasm for its future genetic improvement in the Tibetan Plateau and other regions.

## 2. Materials and Methods

### 2.1. Plant Materials

To characterize genetic diversity in different gene pools, we included three sets of six-rowed naked barley samples, representing the three gene pools that contain (i) the traditional (T), (ii) modern (M), and (iii) germplasm-resources-bank stored (G) varieties. Each set (gene pool) of the samples included 10 varieties and each variety included 30 individuals/plants. Therefore, a total of 900 plants representing the three naked barley gene pools were used in this study. The 10 traditional and 10 modern naked barley varieties were collected from the field in the same season of the year 2019 in Tibet (Table 1). Usually, the fertilizer used for these varieties was 75 kg urea and 112.5 kg diammonium phosphate per hectare (hm^2^). In addition, the 10 germplasm-resources-bank varieties were obtained from the Germplasm Resources Bank (GRB) of the Tibet Academy of Agriculture and Animal Husbandry Sciences in 2019 (Table 1), although these varieties/accessions were collected from the field at different times.

The traditional naked barley varieties were cultivated by the farmers in the hilly areas of Tibet with considerable phenotypic differences among the varieties. The modern naked barley varieties were bred by the Tibet Academy of Agriculture and Animal Husbandry Sciences and are currently cultivated by Tibetan farmers mainly in the basins of the Yarlung Zangbo, Lhasa, and Nyangqu rivers in Tibet. The germplasm-resources-bank stored naked barley varieties were essentially sampled from the traditional varieties that were collected and stored in GRB for about 35 years. The seed germination rates varied between 88 and 95% for the 10 studied T varieties, between 90 and 98% for the 10 studied M varieties, and between 80 and 85% for the 10 studied G varieties, after 5~7 days of seed germination.

### 2.2. DNA Extraction, Amplification, and Genotyping

DNA samples of all studied naked barley varieties were extracted from the leaf tissues of seedlings before the three-leaf stage, although the time for tissue collection varied in a few days. Seedlings were developed from seeds that were germinated and grown in Petri dishes with moist filter papers but without fertilizer application. Seedlings in Petri-dishes were grown in an illuminated incubator (Percival Scientific, Perry, IA, USA) with alternating light/dark (16/8 h) at 25 ± 3 °C. Total genomic DNA was extracted following the modified CTAB protocol [28].

SSR (Simple Sequence Repeats) markers are typically scored as co-dominant markers. This is because SSR markers can simultaneously reveal the presence of both alleles in a heterozygous individual. Thirty-one SSR primer pairs were selected from the cultivated barley (*H. vulgare*) genome, based on the reports in previous studies [29,30,31,32,33,34,35,36,37,38,39,40] with a high level of polymorphisms, to study the genetic diversity of naked barley varieties (Table 2). All the forward primers of the selected SSR primer pairs were labeled with one of the following fluorescent dyes: FAM (blue), HEX (green), ROX (red), and TAMRA (black), respectively [41]. The primer pairs and fluorescent sequences were synthesized by the Sangon Biotech Co., Ltd., (Shanghai, China) (Table 2), and all reagents required for PCR amplifications were produced by the Sangon Biotech Co., Ltd., (Shanghai, China).

The total volume of PCR reaction was 10 μL, including Mg^2+^ 10× PCR buffer 1 μL, 25 mmol·L^−1^ dNTP 0.8 μL, 10 mmol·L^−1^ forward primer 0.04 μL, 10 mmol·L^−1^ reverse primer 0.2 μL, 10 mmol·L^−1^ forward primer fluorescence 0.16 μL, ddH_2_O 5.7 μL, template DNA 2 μL, and 5 U·μL^−1^ Taq DNA Polymerase 0.1 μL. The forward primers were fluorescently labeled to visualize the PCR products by FAM (blue), ROX (red), or HEX (green). The PCR reaction was performed on an ABI 2720 thermal cycler. The reaction program was designed as follows: initial denaturation at 94 °C for 5 min; denaturation at 94 °C for 30 s, annealing at 55 °C for 30 s, extension at 72 °C for 30 s, 30 cycles; final extension at 72 °C for 7 min; and storage at 4 °C. The PCR products were separated and analyzed on a capillary electrophoresis.

For genotyping, the PCR products were separated and analyzed on a capillary electrophoresis analyzer (ABI 3130, Applied Biosystems, Foster, CA, USA). The amplified DNA fragments were scored as genotype (co-dominant) data, according to the size of the SSR fragments for each naked barley sample, owing to the co-dominant feature of the SSR markers [23,29]. The amplified NDA fragments were scored based on fragment length (bp), using the software GeneMapper version 4.1 (Applied Biosystems).

### 2.3. Data Analyses

Genetic Diversity: The following genetic diversity parameters were calculated, including the number of observed alleles per locus (*N_a_*), number of effective alleles per locus (*N_e_*), Shannon information index (*I*), observed heterozygosity (*H_o_*), expected heterozygosity (Nei’s genetic diversity, *H_e_*), percentage of polymorphic loci (*P*), fixation index (*F_st_*) [42], and the estimated gene flow (*Nm*). These genetic diversity parameters were calculated using the software GenAlEx 6.5 [43]. The analysis of molecular variance (AMOVA) was also carried out to estimate the partition of genetic diversity within and among naked barley varieties, at a level of *p* < 0.001 and 9999 permutations [44].

Genetic Structure: The genetic structure of naked barley varieties was analyzed in the Bayesian clustering algorithm-based program STRUCTURE ver. 2.3.4 [45] to visualize the genetic components of the varieties in different gene pools, based on the SSR genotypic data matrix. The running parameters were set as 100,000 burn-in period and 100,000 replicates. The admixture model was selected to analyze the genetic components with the correlated allele frequencies. The number of clusters (*K*-value) was set from 2 to 10 and each *K* value was repeated 10 times, respectively. The Evanno method was used to detect the number of *K* groups that best fit the dataset by the STRUCTURE HARVESTER online program [46]. The software CLUMPP ver. 1.1.2 was used to determine the optimal alignment of the 10 replicates with the ‘Greedy’ algorithm (GREEDY_OPTION = 2, REPEATS = 10,000). The alignment results were visualized using the software DISTRUCT ver. 1.1 [47].

Principal Coordinates Analysis (PCoA): PCoA was undertaken to estimate the genetic differentiation (dissimilarity) of individuals in different naked barley varieties categorized in the three gene pools (T, M, and G). The scatter plot was generated based on the coefficients of the first two principal coordinates to visualize the relationships of naked barley individuals in the traditional (T), modern (M), and germplasm-resources-bank stored (G) varieties using the software GenAlEx ver. 6.5 [43].

Cluster Analyses: The cluster analysis was conducted based on Nei’ s genetic similarity coefficient [48], and the unweighted pair-group method with arithmetic (UPGMA) mean was adopted to construct the similarity output. The UPGMA cluster analysis was conducted with the use of NTSYS-pc software ver. 2.2 [49], based on which the cluster diagram was drawn.

Sampling Strategies and Genetic Diversity: To determine the correlation between the level of genetic diversity and increases in sample sizes within a naked barley variety [50,51], random sampling was conducted to form 5 groups each with 5, 10, 15, 20, 25, to 30 individuals from a barley variety. The average values of genetic diversity parameters (*H_e_*, *I*, *P*) for each group were calculated based on the random draw of each set of samples from a variety for 10 times, using the GenAlEx software ver. 6.5 [43]. The regression of sample sizes (number of individuals) for each group with the average values of genetic diversity was calculated using the SPSS statistics software ver. 28.0.1.0 [52] to generate the best fitting curves based on the regression fitting formula: Ln(*Y*) = *b*_0_ + (*B* − 1/*t*), where *b*_0_, *B* − 1 was the fitting parameter, *t* was the sample sizes, and *Y* was the percentages of the genetic parameters.

## 3. Results

### 3.1. Genetic Diversity of the Tibetan Naked Barley Varieties Included in the Traditional, Modern, and Germplasm-Resources-Bank Stored Gene Pools

A total of 248 alleles based on the 31 selected simple sequence repeat (SSR) loci were detected in the 900 plants/individuals representing the three naked barley gene pools in the traditional (T-1~10), modern (M-1~10), and germplasm-resources-bank stored (G-1~10) varieties from Tibet, China (Table 3). Genetic diversity was estimated based on 30 plants representing one of the 10 varieties from each of the naked barley gene pools. In general, the estimated level of genetic diversity was relatively high for the strictly inbreeding naked barley varieties, although with considerable variations among varieties in the three respective gene pools (Table 3). Interestingly, the level of genetic diversity (estimated by *I* = 1.17 ± 0.09, *H_e_* = 0.58 ± 0.03) in the traditional naked barley varieties (T gene pool) was greater than that (*I* = 0.97 ± 0.07, *H_e_* = 0.50 ± 0.04) in the modern naked barley varieties (M gene pool). However, the level of genetic diversity (*I* = 1.15 ± 0.09, *H_e_* = 0.57 ± 0.04) in the germplasm-resources-bank stored naked barley varieties (G gene pool) was comparable with that in the varieties of the T gene pool (Table 3). Therefore, the richness of genetic diversity in the studied naked barley germplasm was estimated in the following order: traditional (T) > germplasm-resources-bank stored (G) > modern (M) varieties in the three gene pools. Furthermore, much lower observed heterozygosity was detected in the modern varieties compared with that in the traditional and germplasm-resources-bank stored varieties, suggesting a very lower frequency of outcrossing within the modern varieties.

In addition to the estimated richness of genetic diversity, the calculated number of frequent (≥5%) private alleles (denoting the specific/unique alleles in a particular group) was also largely variable among the Tibetan naked barley varieties in the three gene pools (Table 4). Results obtained from the analysis further indicated that a much greater number of private alleles (15 alleles at 7 loci) was detected in the six traditional varieties (T gene pool) than in three modern varieties (M gene pool) that only showed four private alleles at four loci (Table 4). Noticeably, all alleles detected in the modern varieties were completely different from those in the traditional varieties, although a few alleles were similar to those in the germplasm-resources-bank stored varieties (Table 4). The germplasm-resources-bank stored naked barley varieties (G gene pool) showed a comparable number of private alleles although with a slightly reduced number (Table 4).

Interestingly, the level of fixation index, as indicated by the *F_st_* values, was considerably greater in the gene pools of traditional (*F_st_* = 0.61) and germplasm-resources-bank stored varieties (*F_st_* = 0.59) than that (*F_st_* = 0.39) in the modern varieties, suggesting much greater genetic differentiation among the traditional and germplasm-resources-bank stored naked barley varieties. The much lower level of gene flow (*N_m_* = 0.19~0.27) also supported the observation that greater genetic differentiation was formed among the traditional, as well as the germplasm-resources-bank stored naked barley varieties. In contrast, a substantially greater level of gene flow (*N_m_* = 0.85) was detected among the modern naked barley varieties resulting in their much minor genetic differentiation. Given that barley is an autogamous species and that the modern naked barley showed an extremely low level of observed heterozygosity (*H_o_* = 0.01), the detected gene flow is most likely the consequence of seed-mediated gene flow. Altogether, these results further indicated that the traditional naked barley varieties contained abundant genetic diversity with valuable unique alleles, either in the present agroecosystems or in the germplasm-resources-bank storages.

### 3.2. Genetic Structure and Relationships of the Tibetan Naked Barley Varieties Included in the Traditional, Modern, and Germplasm-Resources-Bank Stored Gene Pools

The AMOVA results indicated that a slightly greater portion (~60%) of genetic variation existed among the Tibetan naked barley varieties, including ~7% of genetic variation among the three gene pools, based on the analysis of all 900 samples (Table 5). Consequently, a smaller portion (~40%) of genetic variation was detected within the naked barley varieties (Table 5). However, results based on the analysis of the varieties in each of the three gene pools (T, M, and G) indicated somehow different AMOVA patterns between the traditional/germplasm-resources-bank and modern varieties (Table 5). Noticeably, for the modern naked barley varieties, a slightly greater proportion (~56%) of within-variety genetic variation was detected, resulting in a smaller proportion (~44%) of among-variety genetic variation (Table 5). This finding evidently indicated that considerable genetic variation was presented among the samples or individual plants within a modern naked barley.

To estimate the genetic components of naked barley varieties in the three gene pools (T-1~10, M-1~10, and G-1~10), we conducted the STRUCTURE analysis using the admixture model, based on generating the data matrices of the SSR molecular fingerprints. Results from the STRUCTURE analysis demonstrated similar genetic components of the traditional and germplasm-resources-bank stored naked barley varieties in the T and G gene pools, at the most optimal *K*-value (*K* = 7, mainly represented by the orange, yellow, green, and blue colors) and their neighboring *K*-values (*K* = 6, *K* = 8) (Figure 2). However, the genetic components (mainly represented by the pink and red colors) in the modern naked barley varieties (M gene pool) were considerably different from those in the T and G gene pools (Figure 1). In addition, nearly all naked barley varieties in the T, M, and G gene pools showed a strong admixture of the genetic components, only with very few exceptions in the T and G gene pools, suggesting pollen-mediated gene flow among varieties. All these results clearly indicated that the genetic structure of the traditional and germplasm-resources-bank stored naked barley varieties was relatively similar, but substantially differentiated from the modern naked barley varieties. Noticeably, samples with considerably different genetic components were found within some modern varieties, confirming AMOVA’s results of considerable genetic variation among individual plants within a modern variety.

To understand the overall relationships of the 900 naked barley samples (individuals) represented by the three gene pools in the traditional (T), modern (M), and germplasm-resources-bank stored (G) varieties collected from Tibet of China, we conducted the principal coordinate analysis (PCoA) based on the SSR molecular fingerprints. The PCoA results demonstrated evident genetic variation in the 900 naked barley samples, although with considerable overlaps of the samples among the three gene pools (Figure 3). Obviously, the naked barley samples in the T and G gene pools that were mainly scattered across the positive loads of the first principal coordinate showed a relatively high degree of overlaps, compared with those in the M gene pool that were mainly scattered across the negative loads of the first principal coordinate (Figure 3). Consequently, these results revealed a generally closer genetic relationship of the samples that were included in the traditional and germplasm-resources-bank stored naked barley varieties (T and G gene pools). However, the samples included in the modern naked barley varieties (M gene pool) were somehow genetically deviated from those included in the varieties of T and G gene pools.

To estimate the overall genetic relationships of the 30 naked barley varieties included in the T, M, and G gene pools from Tibet, we conducted the cluster analysis of these varieties, based on the SSR molecular fingerprints. Results generated from the UPGMA cluster analysis indicated that the traditional (T-1~T-9) and germplasm-resources-bank stored (G-1~G-9) varieties were relatively closely linked to each other, although with significant variation as estimated by their genetic similarities (Figure 4). The naked barley varieties in the two (T and G) gene pools presented an obvious intermixed pattern in four groups. However, the modern naked barley varieties (M-1~M-10) were clustered in three groups that were relatively independently separated from most of the varieties in the T and G gene pools. Noticeably, all the modern naked barley varieties were only clustered closely with two traditional (T-8 and T-9) and one germplasm-resources-bank (G-5) varieties (shaded area in Figure 4), suggesting the possible breeding pedigrees of these modern varieties. Altogether, these results confirmed that the Tibetan naked barley varieties included in the T and G gene pools shared relatively closer genetic relationships than the varieties included in the M gene pools.

### 3.3. Relationship Between Variation in Genetic Diversity and Increases in Sample Sizes of Naked Barley Varieties from Different Gene Pools

To determine the most effective sampling strategy for naked barley conservation, we estimated the variation pattern of genetic diversity against the increases in sample sizes (5, 10, 15, 20, 25, and 30 individuals) within varieties from different (T, M, and G) gene pools, using the computer simulation random sampling method. In general, results from the regression analysis showed that genetic diversity within the included naked barley varieties increased quickly with the increases in sample sizes at the first two to three intervals, namely, from 5 to 10 or to 15 individuals (Figure 5). After that, the increases in genetic diversity slowed down, even though the sample sizes continued to increase (Figure 5). Detailed analyses indicated that the average levels (regression fitting curves) of genetic diversity, as measured by Nei’s genetic diversity (*H_e_*) and Shannon information index (*I*), increased rapidly to ~80% of the total levels for all the varieties from the three (T, M, and G) gene pools, when only five individuals were sampled in the analysis (Figure 5). The average genetic diversity (*H_e_* and *I*) rapidly reached the maximum levels when 15 individuals were sampled for different varieties in the three gene pools. However, the average percentage of polymorphic loci (*P*), gradually increased to about 80% of the total levels of genetic diversity when 10 individuals were sampled in the analysis, although with some variation in the G gene pool (Figure 5). The average percentage of polymorphic loci (*P*) slowly reached the maximum levels when more than 20 individuals were sampled from varieties in the three gene pools. These results demonstrated that abundant genetic diversity was harbored by the Tibetan naked barley varieties among individuals both within a variety and among varieties. The generated knowledge could be useful for the design of effective sampling strategies for Tibetan naked barley varieties.

## 4. Discussion

### 4.1. Abundant Genetic Diversity Harbored in Traditional Tibetan Naked Barley Varieties

Our results obtained from the analysis of 900 individuals demonstrate a relatively high level of genetic diversity in the Tibetan naked barley varieties, based on the simple sequence repeat (SSR) molecular fingerprints. As a strictly inbreeding or self-pollinating plant species with clistogamy characteristics [9,51,53], the overall level of genetic diversity, as represented by the Shannon information index (*I* = 1.26 ± 0.09), Nei’s genetic diversity (expected heterozygosity, *H_e_* = 0.60 ± 0.03), and the percentage of polymorphic loci (*P* = 100%), is comparable with many other strictly inbreeding crop species or varieties/lines, such as soybean [51,53], pea [54], emmer and einkorn wheat lines [55], and cultivated barley varieties [9,10,56]. Given that naked barley is a crucial food crop for people living the high-elevation zones, the available knowledge generally indicates that the Tibetan naked barley varieties, with their rich genetic diversity, are extremely valuable for supporting sustainable agriculture and food security programs in the marginal areas. These varieties offer diverse staple food sources that can withstand variable and changing environmental conditions.

In addition, the relatively high level of genetic diversity detected in the Tibetan naked barley varieties, particularly the abundant genetic diversity in the gene pool of traditional varieties (overall *I* = 1.17 ± 0.09; *H_e_* = 0.58 ± 0.03), provides extremely valuable germplasm or genetic resources for the continued conservation and utilization in future breeding programs. This is because the abundant within- and among-variety genetic diversity can be easily utilized in breeding and genetic improvement, not only for naked barley, but also generally for all types of barley varieties [11,18,57]. In fact, many studies indicate that naked barley varieties have played a critical role as genetic resources in barley breeding by donating important genes for tolerance to biotic and abiotic stresses, such as extreme temperature, drought, salinity, plant diseases, and insect pests [16,58,59,60]. These genes are increasingly vital in the context of global climate change, as shifting weather patterns and more frequent extreme weather events exacerbate stressors on crop systems worldwide. By incorporating genes from the traditional naked barley into modern breeding programs, breeders can develop more resilient barley varieties capable of withstanding these challenges. These studies provide excellent examples to justify the critical roles of conserving and effectively utilizing generic resources in crop breeding and genetic improvement, not only to enhance agricultural productivity but also to address the growing need for climate-resilient crops in the face of a warming planet.

Obviously, modern naked barley varieties have undergone significant losses of genetic diversity during the breeding processes. Consequently, the genetic basis of the modern naked barley varieties from the same regions has become relatively narrower than that of the traditional varieties. Facing the extreme environmental conditions exacerbated by global climate change, such as increased frequency of droughts, floods, and shifts in growing seasons, naked barley varieties with broader genetic diversity from different sources will play a critical role in developing modern varieties with wide adaptability [16,24]. Global climate change poses unprecedented challenges to agricultural systems, making it essential to harness the genetic diversity found in crops like Tibetan naked barley. Given that the traditional Tibetan naked barley varieties harbor abundant genetic diversity, their effective exploration and utilization can contribute significantly to the development of resilient barley varieties. This effort will not only benefit Tibet and its neighboring regions, but also have implications for food security and sustainable agriculture in other parts of the world, where the impacts of climate change are becoming increasingly severe.

Detailed analyses of the distribution of genetic diversity in the Tibetan naked barley varieties indicate that the estimated genetic diversity parameters, such as the effective number of alleles, Shannon information index, and Nei’s genetic diversity, are slightly greater in the traditional and germplasm-resources-bank stored varieties than those in the modern varieties. In addition, the level of observed heterozygosity (*H_o_*), which denotes the outcrossing frequency among plants or individuals [61], is twice as high in traditional and germplasm resource bank-stored varieties as in modern varieties. These results suggest that the traditional naked barley varieties have comparatively greater levels of pollen-mediated gene flow through outcrossing, allowing for the long-term accumulation of genetic variation and recombination within the varieties. Consequently, the traditional naked barley varieties hold significant value for future barley breeding.

Coincidentally, the detected fixation indexes or *F_st_* values (0.61 ± 0.03 and 0.59 ± 0.03), among the traditional or germplasm-resources-bank stored naked barley varieties (in T and G gene pools), are much greater than those (*F_st_* = 0.39 ± 0.03) among the modern varieties (in M gene pool). These results suggest considerably greater genetic differentiation among varieties in the traditional and germplasm-resources-bank gene pools than that in the modern gene pool. Moreover, the allelic analysis also indicates that there are many more private alleles with relatively high frequencies (≥0.05) in the traditional and germplasm-resources-bank stored Tibetan naked barley varieties (in T and G gene pools) than those detected in the varieties of modern varieties (M gene pool). The abundant and unique genetic variation identified in the traditional (T) and germplasm-resources-bank stored (G) Tibetan naked barley varieties underscore their potential values as critical genetic resources in future barley breeding.

Altogether, these results demonstrate abundant genetic diversity that is harbored by the traditional Tibetan naked barley varieties and the germplasm-resources-bank stored germplasm, even though only ten traditional or germplasm-resources-bank stored varieties were included in this study. We therefore propose that further extensive exploration, collection, and assessment of traditional naked barley varieties should be undertaken in Tibet under proper guidance, for the effective utilization of the unique genetic resources in future barley breeding programs. Furthermore, given that abundant genetic diversity still exists in naked barley varieties stored at the Tibetan Germplasm Resources Bank, we also propose that more attention should be paid to these stored materials for more intensive and accurate assessment to explore valuable characteristics for their efficient utilization in (naked) barley breeding programs.

### 4.2. Genetic Structure and Relationships of Tibetan Naked Barley Varieties in Different Gene Pools

A better understanding of the distribution and structure of genetic diversity can play an important role in the strategic designs of the efficient conservation and utilization of genetic resources [5,6,62,63]. Such knowledge of genetic diversity is fundamental to guide the proper design of an effective sampling strategy for a particular crop species [23,27,50,51]. Our results in this study evidently indicate the uneven distribution of genetic diversity among the traditional, modern, and germplasm-resources-bank stored Tibetan naked barley varieties. The AMOVA results generally indicate that the major proportion of the genetic variation presented among different naked barley varieties, although the AMOVA patterns are different between the traditional/germplasm-resources-bank varieties and the modern varieties. These results demonstrate the significance of field sampling procedures from different naked barley varieties across a particular region, as a part of the effective conservation strategies [6,25,64]. Such sampling procedures for different varieties can ensure the capture of optimal genetic diversity in the collected naked barley materials. In addition, the AMOVA results also indicate that about 40% of the genetic variation exists within a traditional variety, suggesting that adequate individuals/plants should be collected within a naked barley variety in the field to capture the maximum genetic variation in the conserved genetic resources, as the within-variety sampling strategy. Such a within-variety sampling strategy has been proposed in many other studies of different plant species [50,51]. Noticeably, an exceptionally high proportion of within-variety genetic variation is detected in the modern naked barley varieties. This is most possibly caused by the seed mixture of different varieties/sources by the small-household farmers who are used to exchange modern variety seeds from their neighbors and relatives within a village and between different villages [65]. The frequent exchange of seeds between small-household farmers is also reported in the remote rice ecosystems in China [66]. The unusually high level of migrations (*N_m_*) or seed-mediated gene flow among the modern naked barley varieties can also explain the remarkably high proportion of seed mixture in the modern varieties as detected in this study (Table 3 and Figure 2) for such a strictly inbreeding crop species. The similar situation was also observed between weedy rice populations, which promoted the within-population genetic diversity [61].

Our results based on the STRUCTURE analysis indicate relatively different genetic components among naked barley varieties included in the traditional and germplasm-resources-bank gene pools, at the most optimal K value (K = 7) and the neighboring K values. Both traditional and germplasm-resources-bank stored varieties showed an evident admixture pattern of the genetic structures. These results suggest that abundant genetic diversity exists in different traditional and germplasm-resources-bank stored naked barley varieties [31,32,37]. Consequently, these varieties are indeed very important for serving as genetic resources for sustainable uses in barley genetic improvement, apart from their great values as food sources [2,4,25]. It is also evident that the traditional and germplasm-resources-bank stored naked barley varieties share similar genetic components. This finding suggests that the germplasm-resources-bank stored naked barley varieties were collected essentially based on the traditional naked barley varieties, although there are some differences in their genetic components after static storage in the germplasm-resources-bank for about 30 years. Probably, the continued evolution of the traditional barley varieties in situ has caused such changes between the two gene pools, as found in other crop species [62,63]. Therefore, it is very important to carry out a more detailed pair-wise comparison between field-collected and germplasm-resources-bank stored naked barley varieties from the same locations to identify the changes in their genetic components using molecular fingerprinting methods. The knowledge generated as such will provide deep insights for understanding the roles of adaptive evolution of naked barley in the changing agroecosystems, which will help us to design effective in situ or on-farm conservation of naked barley germplasm, particularly in mountainous Tibet.

Yet, the modern naked barley varieties demonstrate a somewhat deviated genetic structure with different components from the traditional and germplasm-resources-bank stored naked barley varieties, although these varieties still share some degrees of genetic identity. These results indicate that the modern naked barley varieties are selected and produced most likely only from a limited number of traditional naked barley varieties in the breeding programs. Noticeably, quite different genetic components were detected among some of the samples/individuals in the modern naked barley varieties, clearly suggesting the mixture of individuals with different genetic backgrounds in the varieties. This finding is highly agreeable with that indicated by the genetic diversity and AMOVA patterns of modern varieties, in which a high proportion of within-variety variation (56%) and among-variety gene flow (*N_m_* = 0.85) are detected. Such an unexpected pattern for a strictly inbreeding plant species can only be reasonably explained by the accidental mixture of seeds between different naked barley varieties through farmers’ seed exchange, as proposed previously in this study. In field practices, Tibetan farmers often complain about the degradation of naked barley varieties after cultivation for a few years (authors’ field observations). Therefore, an appropriate cultivation protocol or procedure should be established and used as a guide for farmers in field practices [65] to avoid the rapid degradation of their newly bred modern naked barley varieties.

In this study, our PCoA and cluster analyses strongly support the above conclusions that modern naked barley varieties somehow deviate from the traditional and germplasm-resources-bank stored naked barley varieties. It is evident that the modern varieties only cover a small fraction of the genetic diversity contained in the traditional and germplasm-resources-bank stored varieties. Particularly, our results from the cluster analysis clearly indicate that only two traditional and one germplasm-resources-bank stored naked barley varieties (Figure 4) have an intimate linkage with all the modern varieties; in other words, a limited number of traditional varieties (as parents) is used in the breeding of modern varieties. Most likely, the modern naked barley varieties were produced by only including a very small portion of genetic variation in traditional varieties and having a considerably narrow genetic background, which usually poses great challenges for these modern varieties in the present and the future. For example, limited genetic diversity may lead to reduced resistance to diseases and insect pests, reduced sustainability for long-term utilization, and lower adaptability to environmental changes. In fact, we have observed in the field expeditions that some of the modern varieties cannot produce seeds normally at an elevation higher than 4500 m in the Shigatse District of Tibet, simply because of the poor adaptation to the insufficient cumulative temperature. In contrast, the traditional naked barley varieties grown in the same locations can produce seeds normally with guaranteed yield for these varieties. Therefore, we strongly propose to largely broaden or expand the genetic bases of modern naked barley varieties in breeding programs by applying diverse germplasm from different sources [67]. Such breeding efforts will greatly increase the adaptability of modern naked barley varieties to different environmental conditions. This is essential for the sustainable production and development of these varieties in regions that are characterized by considerably variable environmental conditions, particularly in these regions under the snorers of drastic climate changes.

### 4.3. Implications of the Generated Knowledge in Sampling Strategy for Naked-Barley Germplasm Conservation and Breeding Programs

The effective conservation of genetic resources or germplasm of various crop species is essential for human beings to sustainably utilize and develop crop varieties of these species in diversified agroecosystems [2,7,26]. Likewise, the effective conservation of genetic resources of such a unique crop as naked barley in Tibet—a particular agroecosystem with great geographical and biological diversity, is also extremely important for people living in these marginal areas. To achieve this goal, it is necessary to design strategies to effectively conserve the naked barley varieties using efficient sampling methods in the field across the agroecosystems. Knowledge generated from this study has important implications to guide the conservation of naked barley germplasm with proper sampling strategies, as well as in the breeding programs for this crop [21,27,39].

First, our findings based on the obtained results from this study indicated that it is greatly valuable to collect, evaluate, and utilize genetic resources continually in the Tibetan naked barley varieties, particularly those categorized in the traditional gene pool. In general, our results showed a considerably greater level of genetic diversity as estimated by the overall genetic parameters. Particularly, the traditional naked barley varieties contain far greater numbers of private or unique alleles than the modern varieties, which will contribute more significantly to the further genetic improvement of modern varieties. Continued collection of more varieties in the traditional gene pool, regardless of whether these varieties or names are represented in the germplasm-resources-bank storage, should be the prioritized task for the conservation of naked barley in Tibet. Second, given that the distribution of genetic diversity is mainly among different traditional naked barley varieties, the collection of germplasm samples should consider including more varieties across a given region, instead of concentrating on only one variety or a few varieties in a particular region. Such a collecting strategy can capture the most possible genetic diversity harbored in the traditional naked barley varieties.

In addition, the results of the correlation analysis demonstrate the relationship between the increases in genetic diversity (0~100%) and the sizes of samples/individuals (5~30) drawn randomly from each of the traditional, modern, and germplasm-resources-bank stored naked barley varieties. The obtained correlation patterns provide a valuable guide for germplasm sampling of naked barley varieties in the field. The sampling size or the number of plants/individuals to be collected from a naked barley field can be determined following the correlation curves revealed in this study. Generally, the correlation patterns suggest that to sample five plants/individuals within a naked barley field, either within the traditional or modern gene pool, about 80% of total genetic diversity (as represented by *H_e_* and *I*) can be captured. In addition, close to 100% of total genetic diversity can be captured with a naked barley field, if the sample size is increased to about ten plants/individuals (Figure 5). Therefore, we propose based on the findings from this study that the proper sample size of about 5~10 plants/individuals from a variety should be considered when collecting in a naked barley field. Such a field practice will ensure the capture of the optimal genetic diversity for the effective conservation of naked barley germplasm. Furthermore, our correlation curves also indicate that a greater sample size should be considered in the field collection, if we attempt to include more polymorphic loci in the collected samples from naked barley varieties (Figure 5). Obviously, the detailed study of genetic diversity, including its richness, distribution, and genetic structure, can provide important information to guide the effective conservation of naked barley genetic resources, apart from efficient utilization in future breeding.

## 5. Conclusions

Our results clearly demonstrate that Tibetan naked barley varieties harbor abundant genetic diversity, although genetic diversity is not evenly distributed across the traditional, modern, and germplasm-resources-bank stored varieties. Both the principal coordinates and STRUCTURE analyses indicate substantial deviation of the modern varieties from the traditional and germplasm-resources-bank stored varieties. The cluster analysis suggests the relatively narrow genetic background of modern varieties, likely due to the limited use of traditional and germplasm-resources-bank varieties in modern breeding programs. Therefore, we strongly recommend expanding the genetic base of modern naked barley varieties by incorporating diverse germplasm from different sources. Correlations between the average genetic diversity levels and sample sizes (number of samples) within varieties suggest sampling strategies for the effective collection of naked barley genetic resources in the field. The knowledge generated from this study has important implications not only for the sustainable utilization of naked barley as a food source, but also for its effective conservation as germplasm in barley breeding programs, both in Tibet and in other regions around the world.

## Figures and Tables

**Figure 1 biology-13-01018-f001:**
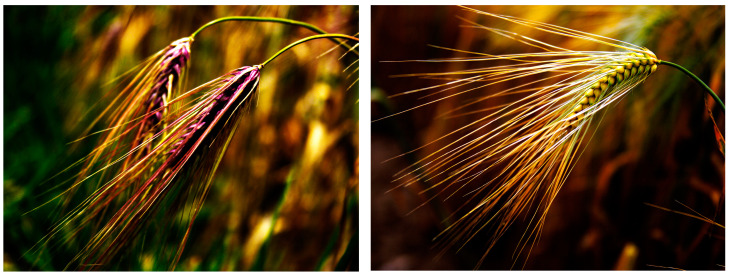
Panicles of two naked barley varieties from the Qinghai-Tibet Plateau, with the traditional variety (Nienachareng) showing on the left and the modern variety (Zangqing 2000) on the right.

**Figure 2 biology-13-01018-f002:**
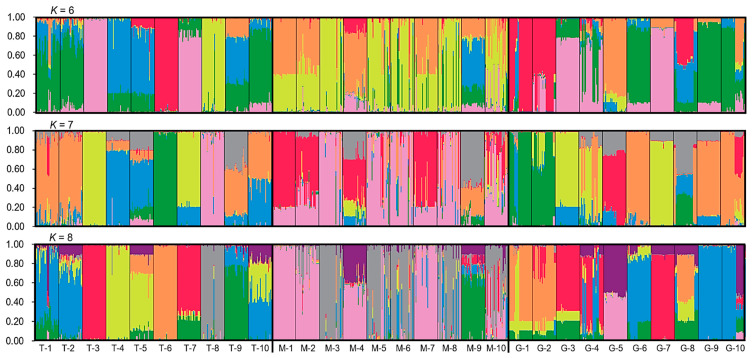
Bar plots indicating different genetic components of the 900 samples representing traditional (T-1~10), modern (M-1~10), and germplasm-resources-bank stored (G-1~10) naked barley varieties, based on the STRUCTURE analysis of SSR markers at the most optimal *K*-value (*K* = 7) and their neighboring *K*-values. Each sample is represented by a single vertical line, proportional to different genetic components. Many samples showed an admixture of genetic components. Different colors represent different genetic components.

**Figure 3 biology-13-01018-f003:**
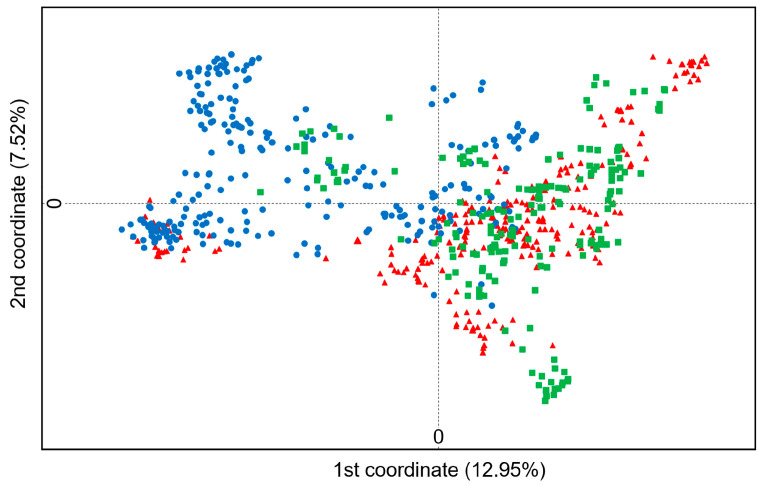
Scatterplots demonstrating genetic relationships of 900 samples representing traditional (red triangles), modern (blue dots), and germplasm-resources-bank stored (green squares) naked barley varieties from Tibet of China, based on the principal coordinate analysis of SSR markers.

**Figure 4 biology-13-01018-f004:**
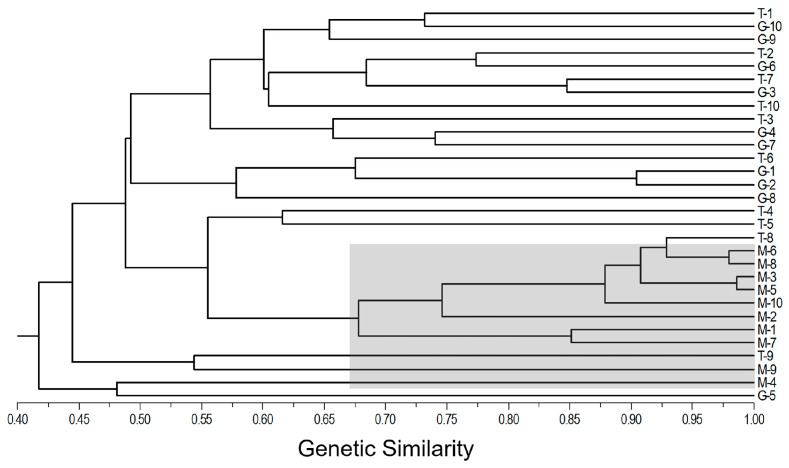
UPGMA dendrogram illustrating genetic relationships of traditional (T), modern (M), and germplasm-resources-bank stored (G) naked barley varieties (1–10) from Tibet of China, based on analyses of the pairwise Nei’s genetic similarity of SSR markers. Shaded clusters indicate the M naked barley varieties (M-1~M10) that are related to T-8, T-9, and G-5.

**Figure 5 biology-13-01018-f005:**
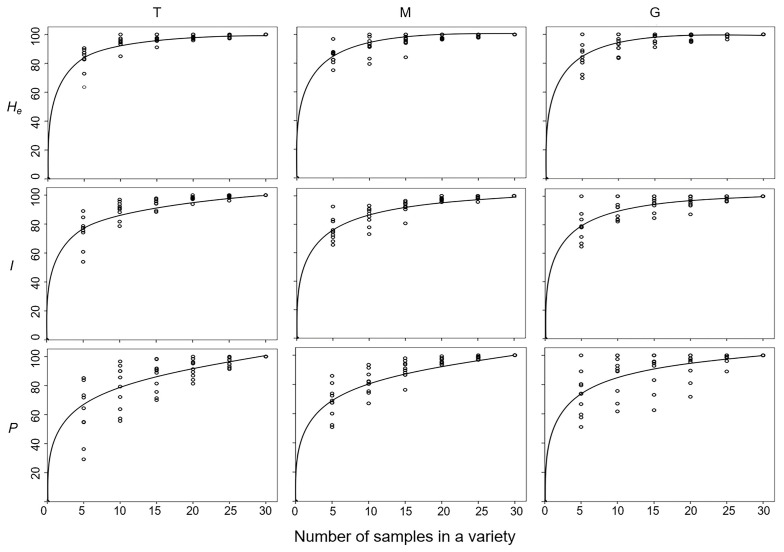
The charts generated based on the regression analysis illustrating increases in genetic diversity (0~100%) with the increased number of samples/individuals from 5~30 intervals in the traditional (T), modern (M), and germplasm-resources-bank stored (G) naked barley varieties. Genetic diversity was represented by the parameters of Nei’s genetic diversity (*H_e_*), Shannon information index (*I*), and the percentage of polymorphic loci (*P*). The lines indicate the regression fitting curves; the rings represent the average values calculated from each of the 10 varieties in the T, M, and G gene pools.

**Table 1 biology-13-01018-t001:** Names of the naked barley varieties from traditional (T-1~10), modern (M-1~10), and germplasm-resources-bank stored (G-1~10) gene pools used in this study with information on their collecting sites and locations (longitude, latitude, and altitude) in Tibet, China.

Variety Name	Variety Type	Variety Code	Collection Site	Longitude (N)	Latitude (E)	Altitude (m)
Chanima	Traditional	T-10	Medro Gongkar, Lhasa	29°33′	92°20′	3989
Chujuma	Traditional	T-1	Khangmar, Shigatse	28°27′	89°40′	4434
Gamuguoduo	Traditional	T-9	Rutok, Ngari	33°42′	79°65′	4263
Gamuguori	Traditional	T-8	Samdruptse, Shigatse	29°10′	90°43′	3881
Jiaqiong	Traditional	T-3	Chosum, Lhoka	29°07′	92°19′	3830
Niegachatong	Traditional	T-5	Dayak, Chamdo	29°85′	96°70′	3659
Niegenguoluo	Traditional	T-7	Medro Gongkar, Lhasa	29°74′	91°97′	3843
Nienachareng	Traditional	T-6	Lhodak, Lhoka	28°39′	90°79′	3917
Nienachatong	Traditional	T-4	Lhodak, Lhoka	27°99′	91°94′	3853
Wodun	Traditional	T-2	Gampa, Shigatse	28°46′	88°63′	4574
5171	Modern	M-8	Panam, Shigatse	29°22′	90°46′	3539
Shandong 18	Modern	M-4	Nyingri, Nyingtri	29°51′	94°64′	3037
Shanqing 9	Modern	M-2	Nedong, Lhoka	29°21′	91°83′	3556
Sulaqing 2	Modern	M-6	Taktse, Lhasa	29°42′	91°26′	3692
Xila 22	Modern	M-1	Chushur, Lhasa	29°23′	90°51′	3431
Xila 23	Modern	M-9	Panam, Shigatse	28°76′	89°14′	4120
Zangqing 17	Modern	M-10	Chushur, Lhasa	29°38′	90°90′	3598
Zangqing 2000	Modern	M-5	Nedong, Lhoka	29°04′	91°86′	3763
Zangqing 320	Modern	M-3	Nyemo, Lhasa	29°29′	90°16′	3915
Zangqing 85	Modern	M-7	Nyemo, Lhasa	29°26′	90°10′	3815
Chanima	Germplasm-bank	G-4	ZDM06468	29°83′	91°73′	3823
Chujuma	Germplasm-bank	G-10	ZDM06775	29°65′	94°36′	2941
Gamuguoduo	Germplasm-bank	G-8	ZDM06088	30°30′	81°17′	3900
Jiaqiong	Germplasm-bank	G-7	ZDM06615	29°07′	92°20′	3985
Niegachareng	Germplasm-bank	G-6	ZDM07653	29°26′	88°88′	3842
Niegachatong	Germplasm-bank	G-5	ZDM05516	29°67′	97°84′	3806
Niegenguoluo	Germplasm-bank	G-3	ZDM06261	29°27′	92°02′	3687
Nienachareng	Germplasm-bank	G-1	ZDM07183	29°03′	91°68′	3896
Nienachareng	Germplasm-bank	G-2	ZDM05547	28°38′	90°87′	3842
Wodun	Germplasm-bank	G-9	ZDM06907	28°28′	88°52′	4375

**Table 2 biology-13-01018-t002:** The *Hordeum vulgare* SSR loci and their primer-pair sequences, motifs, and locations on the chromosomes used in this study.

SSR Locus [References]	Primer-Pair Sequence (5′-3′)		Motif (No. of Repeats)	No. Chromosome (L = Long Arm, S = Short Arm)
	Forward Primer (5′)	Reverse Primer (3′)		
Bmac0032 [29]	CCATCAAAGTCCGGCTAG	GTCGGGCCTCATACTGAC	AC (7) T (CA) (15) AT (9)	1 HL
Bmac0064 [30]	CTGCAGGTTTCAGGAAGG	AGATGCCCGCAAAGAGTT	CA (8)	7 HL
Bmac0090 [30]	CCGCACATAGTGGTTACATC	ACATCAACCCTCCTGCTC	AC (20)	1 HL
Bmac0096 [30]	GCTATGGCGTACTATGTATGGTTG	TCACGATGAGGTATGATCAAAGA	AT (6) AC (18)	5 HS
Bmac0163 [33]	TTTCCAACAGAGGGTATTTACG	GCAAAGCCCATGATACATACA	AC (6) GC (3) AC (17)	5 HS
Bmac0316 [29]	ATGGTAGAGGTCCCAACTG	ATCACTGCTGTGCCTAGC	AC (19)	6 HS
Bmac0389 [34]	TATGATTGCACGTCCGTTGT	AGGTTTTGATGCCTTGTTGG	TATC (5)	1 HS
Bmag0009 [29]	AAGTGAAGCAAGCAAACAAACA	ATCCTTCCATATTTTGATTAGGCA	AG (13)	6 HL
Bmag0011 [30]	ACAAAAACACCGCAAAGAAGA	GCTAGTACCTAGATGACCCCC	AG (13) AG (10) GA (7)	7 HL
Bmag0120 [32]	ATTTCATCCCAAAGGAGAC	GTCACATAGACAGTTGTCTTCC	AG (15)	7 HL
Bmag0211 [29]	ATTCATCGATCTTGTATTAGTCC	ACATCATGTCGATCAAAGC	CT (16)	1 HS
Bmag0223 [29]	TTAGTCACCCTCAACGGT	CCCCTAACTGCTGTGATG	AG (16)	5 HL
Bmag0225 [29]	AACACACCAAAAATATTACATCA	CGAGTAGTTCCCACGTGAC	AG (26)	7 HL
Bmag0323 [35]	TGACAAACAAATAATCACAGG	TTTGTGACATCTCAAGAACAC	CT (24)	5 HL
Bmag0337 [35]	ACAAAGAGGGAGTAGTACGC	GACCCATGATATATGAAGATCA	AG (22)	5 HL
Bmag0558 [36]	TCAAATTCAGTTGCTGCTGG	CTCCTACCTATCTCGGCGTG	ACAT (8)	3 HS
Bmag0603 [30]	ATACCATGATACATCACATCG	GGGGGTATGTACGACTAACTA	AG (24)	3 HL
Bmag0693 [34]	AGTTGAGTTATCTGGGAGCA	AAACCCTAGGGCACCGACCT	TA (5)	2 HL
Bmag0759 [34]	CTCCATGACGATGAGGAGAAG	AAGAACACCATATGATCCAAC	GA (12)	6 HL
Bmag0835 [34]	CTTATGTCCGGGGACTTCCT	TGTTGCTGGAGCAAGAAGAA	GA (20)	5 HS
Bmag0870 [30]	AACCATAGGATTTGTACTAGTTTC	TCATGACATCTCAAGAACG	TC (8) CT (8) CT (6)	6 HL
GBM1215 [36]	ATGACCAGAAAACGCCTGTC	GGATTCTGCACACACGAGAA	AC (10)	6 HS
GMS0027 [32]	CTTTTTCTTTGACGATGCACC	TGAGTTTGTGAGAACTGGATGG	GT (5) CT (2) GT (27)	5 HL
HVM20 [31]	CTCCACGAATCTCTGCACAA	CACCGCCTCCTCTTTCAC	GA (19)	1 HL
HVM36 [29]	TCCAGCCGAACAATTTCTTG	AGTACTCCGACACCACGTCC	GA (13)	2 HS
HVM43 [37]	GGATTTTCTCAAGAACACTT	GCGTGAGTGCATAACATT	CA (9)	1 HS
SCSSR02748 [38]	GGTGCATTTGGAAGTCTAGG	ATAGCAAGTGCCAAGTGAGC	CT (11)	1 HL
SCSSR05599 [39]	TTCCATCATAACAGCAATGG	TTCGTCGAAGGCTATGTAGG	ACA (8)	6 HL
SCSSR07970 [40]	TGCATTGGGAGTGCTAGG	TGCAAGAAGCCAAGAATACC	TGC (5)	7 HS
SCSSR09398 [39]	AGAGCGCAAGTTACCAAGC	GTGCACCTCAGCGAAAGG	GAA (10)	6 HS
SCSSR10148 [29]	AAGCAGCAAAGCAAAGTACC	TCATCAGCATCTGATCATCC	GT (10)	5 HL

**Table 3 biology-13-01018-t003:** Genetic diversity parameters in the three gene pools of the traditional (T1~10), modern (M-1~10), and germplasm-resources-bank stored (G-1~10) naked barley varieties collected in Tibet, China.

Variety Code	*N **	*N_a_*	*N_e_*	*I*	*H_o_*	*H_e_*	*P*	*F_st_*	*N_m_*
T-1	30	3.26 ± 0.32	2.02 ± 0.15	0.76 ± 0.08	0.02 ± 0.02	0.42 ± 0.04	87.10%		
T-2	30	3.19 ± 0.29	2.08 ± 0.19	0.74 ± 0.09	0.02 ± 0.02	0.41 ± 0.04	87.10%		
T-3	30	1.52 ± 0.17	1.20 ± 0.07	0.18 ± 0.06	0.02 ± 0.02	0.10 ± 0.03	29.03%		
T-4	30	1.71 ± 0.15	1.15 ± 0.05	0.18 ± 0.04	0.03 ± 0.03	0.10 ± 0.03	51.61%		
T-5	30	1.87 ± 0.20	1.23 ± 0.08	0.22 ± 0.06	0.03 ± 0.03	0.12 ± 0.04	48.39%		
T-6	30	1.61 ± 0.16	1.23 ± 0.09	0.20 ± 0.06	0.03 ± 0.02	0.12 ± 0.04	45.16%		
T-7	30	1.81 ± 0.19	1.48 ± 0.11	0.34 ± 0.07	0.02 ± 0.02	0.22 ± 0.05	51.61%		
T-8	30	2.26 ± 0.19	1.27 ± 0.08	0.29 ± 0.05	0.02 ± 0.02	0.16 ± 0.03	80.65%		
T-9	30	2.90 ± 0.26	1.85 ± 0.14	0.64 ± 0.08	0.02 ± 0.02	0.37 ± 0.04	93.55%		
T-10	30	2.39 ± 0.19	1.50 ± 0.10	0.46 ± 0.06	0.02 ± 0.02	0.27 ± 0.04	80.65%		
T-average	30	2.25 ± 0.21	1.50 ± 0.11	0.40 ± 0.07	0.02 ± 0.02	0.23 ± 0.04	65.49%		
T-Overall	300	6.32 ± 0.68	2.99 ± 0.28	1.17 ± 0.09	0.02 ± 0.02	0.58 ± 0.03	100%	0.61 ± 0.03	0.19 ± 0.03
M-1	30	1.81 ± 0.17	1.34 ± 0.08	0.30 ± 0.06	0.01 ± 0.01	0.19 ± 0.04	54.84%		
M-2	30	2.52 ± 0.19	1.58 ± 0.11	0.50 ± 0.06	0.01 ± 0.01	0.30 ± 0.40	87.10%		
M-3	30	2.55 ± 0.19	1.40 ± 0.08	0.42 ± 0.05	0.004 ± 0.004	0.24 ± 0.03	87.10%		
M-4	30	2.45 ± 0.22	1.59 ± 0.12	0.50 ± 0.07	0.01 ± 0.01	0.29 ± 0.04	90.32%		
M-5	30	3.42 ± 0.24	1.66 ± 0.10	0.65 ± 0.06	0.01 ± 0.01	0.34 ± 0.03	93.55%		
M-6	30	2.97 ± 0.22	1.51 ± 0.10	0.52 ± 0.06	0.005 ± 0.004	0.28 ± 0.04	90.32%		
M-7	30	2.39 ± 0.20	1.29 ± 0.06	0.35 ± 0.05	0.001 ± 0.001	0.19 ± 0.03	83.87%		
M-8	30	2.90 ± 0.23	1.70 ± 0.11	0.61 ± 0.07	0.000 ± 0.000	0.34 ± 0.04	90.32%		
M-9	30	2.00 ± 0.15	1.47 ± 0.10	0.41 ± 0.07	0.000 ± 0.000	0.24 ± 0.04	70.97%		
M-10	30	3.13 ± 0.21	2.01 ± 0.11	0.77 ± 0.06	0.001 ± 0.001	0.45 ± 0.03	93.55%		
M-average	30	2.61 ± 020	1.34 ± 0.10	0.50 ± 0.06	0.01 ± 0.01	0.29 ± 0.07	84.19%		
M-Overall	300	5.55 ± 0.41	2.28 ± 0.15	0.97 ± 0.07	0.005 ± 0.004	0.50 ± 0.04	100%	0.39 ± 0.03	0.85 ± 0.27
G-1	30	2.07 ± 0.15	1.39 ± 0.07	0.40 ± 0.06	0.01 ± 0.01	0.23 ± 0.03	70.97%		
G-2	30	1.87 ± 0.15	1.55 ± 0.09	0.42 ± 0.06	0.01 ± 0.004	0.28 ± 0.04	64.52%		
G-3	30	1.42 ± 0.13	1.15 ± 0.07	0.13 ± 0.05	0.02 ± 0.02	0.08 ± 0.03	32.26%		
G-4	30	2.26 ± 0.20	1.72 ± 0.12	0.56 ± 0.07	0.01 ± 0.01	0.34 ± 0.04	74.19%		
G-5	30	2.30 ± 0.20	1.69 ± 0.12	0.50 ± 0.08	0.01 ± 0.01	0.31 ± 0.05	67.74%		
G-6	30	2.45 ± 0.26	1.49 ± 0.13	0.43 ± 0.07	0.02 ± 0.02	0.24 ± 0.04	74.19%		
G-7	30	1.52 ± 0.14	1.19 ± 0.07	0.18 ± 0.05	0.02 ± 0.02	0.10 ± 0.03	38.71%		
G-8	30	2.16 ± 0.16	1.60 ± 0.09	0.51 ± 0.07	0.02 ± 0.02	0.31 ± 0.04	77.42%		
G-9	30	1.61 ± 0.18	1.23 ± 0.07	0.21 ± 0.06	0.02 ± 0.02	0.12 ± 0.04	38.71%		
G-10	30	2.36 ± 0.15	1.78 ± 0.09	0.61 ± 0.06	0.02 ± 0.02	0.38 ± 0.04	87.10%		
G-average	30	2.00 ± 0.17	1.48 ± 0.09	0.40 ± 0.06	0.02 ± 0.02	0.24 ± 0.04	62.58%		
G-Overall	300	5.55 ± 0.43	3.06 ± 0.31	1.15 ± 0.09	0.02 ± 0.02	0.57 ± 0.04	100%	0.59 ± 0.03	0.27 ± 0.08
Overall	900	8.00 ± 0.70	3.13 ± 0.28	1.26 ± 0.09	0.02 ± 0.01	0.60 ± 0.03	100%	0.59 ± 0.02	0.20 ± 0.02

* *N*, number of samples; *N_a_*, number of alleles per locus; *N_e_*, number of effective alleles per locus; *I*, Shannon-Wiener index; *H_o_*, observed heterozygosity; *H_e_*, Nei’s genetic diversity (expected heterozygosity) (Nei, 1978) [48]; *P*, percentage of polymorphism loci; *F_st_*, fixation index (Wright, 1951) [42]; *N_m_*, number of migrations (gene flow) per generation.

**Table 4 biology-13-01018-t004:** Private alleles (with frequency ≥ 0.05) detected in the traditional (T), modern (G), and germplasm-resources-bank stored (G) naked-barley varieties from Tibet, China.

Variety Code	SSR Locus	Allele	Frequency	Variety Code	SSR Locus	Allele	Frequency	Variety Code	SSR Locus	Allele	Frequency
T-7	Bmag0639	314	0.57	M-4	Bmag0759	205	0.97	G-5	Bmag0835	253	0.50
T-1	Bmac0032	279	0.20	M-4	Bmag0011	192	0.37	G-8	Bmac0032	247	0.70
T-6	Bmac0032	271	0.60	M-10	Bmag0009	180	0.07	G-8	Bmag0693	239	0.27
T-9	Bmac0032	265	0.17	M-2	HVM36	131	0.07	G-8	Bmag0211	211	0.70
T-9	Bmac0032	263	0.37					G-6	Bmag0759	201	0.17
T-9	Bmac0090	240	0.07					G-6	Bmac0096	185	0.07
T-6	SCSSR09398	232	0.92					G-2	Bmag0323	169	0.07
T-2	SCSSR09398	228	0.05					G-5	Bmag0011	158	0.33
T-9	Bmac0389	217	0.10					G-8	Bmag0603	140	0.10
T-3	SCSSR09398	192	0.60					G-6	Bmag0223	137	0.27
T-3	SCSSR09398	190	0.35					G-9	Bmag0870	129	0.80
T-9	Bmag0870	153	0.07								
T-1	Bmag0337	152	0.10								
T-9	Bmag0603	138	0.20								
T-9	Bmag0603	136	0.37								

**Table 5 biology-13-01018-t005:** Analysis of molecular variance (AMOVA) of 30 naked barley varieties in the traditional (T), modern (M), and germplasm-resources-bank stored (G) gene pools showing the partition of genetic diversity within and among gene pools. *p*-value (<0.01) estimates are based on 99,999 permutations.

Source	*d.f.* *	*SS*	*MS*	*Est. var.*	%
Within varieties	290	4017.70	13.85	13.85	37
Among varieties	9	6592.89	732.54	23.96	63
Subtotal (T-gene pool)	299	10,610.59	- **	37.81	100
Within varieties	290	5257.90	18.13	18.13	56
Among varieties	9	3989.74	443.30	14.17	44
Subtotal (M-gene pool)	299	9247.64	-	32.30	100
Within varieties	290	4298.67	14.82	14.82	40
Among varieties	9	6211.86	690.21	22.51	60
Subtotal (G-gene pool)	299	10,510.52	-	37.34	100
Within varieties	870	13,570.60	15.60	15.60	40
Among varieties	27	16,795.88	622.07	20.22	53
Among gene pools	2	2856.53	1428.26	2.69	7
Total	899	33,223.01	2065.93	38.50	100

* *d.f.*, degree of freedom; *SS*, sum of squared deviations; *MS*, mean of squared deviations; *Est. var.*, variance component estimates; %, total percentage of total variation. ** - indicates no values within a gene pool.

## Data Availability

All the data were shown in the article.

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
