# Peer review of "Abundant Genetic Diversity Harbored by Traditional Naked Barley Varieties on Tibetan Plateau: Implications in Their Effective Conservation and Utilization"

_biology, 2024, doi:10.3390/biology13121018_

Round 1

Reviewer 1 Report

Comments and Suggestions for Authors

Thanks to the authors for submitting this study titled "Abundant genetic diversity harbored by traditional naked barley varieties on Tibetan Plateau: Implications in their effective conservation and utilization"

The study is good and useful but there are some comments please respond to them or modify as required in order to improve the manuscript.

·       Since the tolerance of bare barley to low irrigation water conditions has been mentioned, it is worth adding its tolerance to irrigation water salinity.

·       Also add a paragraph to the introduction about the appropriate fertilization rates and sowing rate for the studied barley, citing appropriate references.

·       In lines 101-114: Samples of 30 genotypes were collected from the study area, it would be better to also mention that barley varieties were taken from the germplasm-resources-bank.

·       Were samples taken for all varieties from all studied areas in the same season?

·       Was the germination percentage of the seeds estimated? Or the germination speed? Because this is important in evaluating and distinguishing between genotypes.

·       What growth medium was used when germinating the seeds?

·       Did all the planted varieties reach the 3-leaf stage at the same time?

·       Can you shorten the title of figures 2, 3 and 4?

·       Line 410: It is best to Check the references cited, and shorten them to 2 or 3, that is closest to the topic being discussed and also the references on lines 454 and 553.

·       Appropriate references should be cited on several topics mentioned in lines (480-540).

Author Response

POINT-BY-POINT RESPONSES TO REVIEWERS’ COMMENTS

POINT-BY-POINT RESPONSES TO REVIEWER #1

General comments to the Authors

Thanks to the authors for submitting this study titled "Abundant genetic diversity harbored by traditional naked barley varieties on Tibetan Plateau: Implications in their effective conservation and utilization." The study is good and useful but there are some comments please respond to them or modify as required in order to improve the manuscript.

Responses – We greatly appreciate the constructive comments to our manuscript, and therefore, we have modified our manuscript following these comments.

Comment-1: Since the tolerance of bare barley to low irrigation water conditions has been mentioned, it is worth adding its tolerance to irrigation water salinity.

Responses – Thanks for this comment. We have added relevant information regarding naked barley’s tolerance to drought and salinity in the revised manuscript with appropriate references. (Please see line 433-434).

Comment-2: Also add a paragraph to the introduction about the appropriate fertilization rates and sowing rate for the studied barley, citing appropriate references.

Responses – Thanks for the comment. We have added relevant information about fertilization and sowing rates in the revised manuscript (Please see line 129-131).

Comment-3: In lines 101-114: Samples of 30 genotypes were collected from the study area, it would be better to also mention that barley varieties were taken from the germplasm-resources-bank.

Responses – Thanks for this useful comment and we have added such information in Introduction (Please see line 111-114) Materials and Methods of the revised manuscript. (Please see line 132-137).

Comment-4: Were samples taken for all varieties from all studied areas in the same season?

Responses – We appreciate this point and have added such information in Materials and Methods of the revised manuscript. In fact, the traditional and modern varieties were collected in the same season in year 2019. In addition, the materials from germplasm-resources-bank were also requested in 2019. (Please see line 132-137).

Comment-5: Was the germination percentage of the seeds estimated? Or the germination speed? Because this is important in evaluating and distinguishing between genotypes.

Responses – Thanks for the suggestion. Accordingly, we have added some information about seed germination rates for traditional, modern, and germplasm-resources-bank varieties in Materials and Methods of the revised manuscript. (Please see line 145-147).

Comment-6: What growth medium was used when germinating the seeds?

Responses – Thanks for the comment. All naked barley seeds were germinated and grown in petri-dishes with moist filter papers without any fertilizer. We have added a few sentences to further elaborate this in the revised manuscript. (Please see line 146-150).

Comment-7: Did all the planted varieties reach the 3-leaf stage at the same time?

Responses – DNA samples were extracted from leaf-tissues of seedlings before the three-leaf-stage, although the time for tissue collection varied in a few days. We have added mentioned this in the revised manuscript. (Please see line 149-153).

Comment-8: Can you shorten the title of figures 2, 3 and 4?

Responses – Thanks a lot for the suggestions and we have tried to make the figure legends shorter but to keep sufficient explanations.

Comment-9: Line 410: It is best to Check the references cited, and shorten them to 2 or 3, that is closest to the topic being discussed and also the references on lines 454 and 553.

Responses – We have checked the references and only kept the most relevant ones accordingly.

Comment-10: Appropriate references should be cited on several topics mentioned in lines (480-540).

Responses – We have checked the references and only kept the most relevant ones accordingly.

Reviewer 2 Report

Comments and Suggestions for Authors

- Up-to-date references should be used.

- DOI numbers should be included whenever possible (line 186-187 of the writing guidelines).

- This paper makes important contributions to genetic diversity analysis and conservation strategies. However, issues such as genetic problems of modern varieties and climate change preparedness should be emphasized more.

- Concrete suggestions for expanding the genetic base of modern varieties could be added.

- There could be broader discussions in the context of climate change.

- The language could be slightly simplified.

Comments on the Quality of English Language

- The language could be slightly simplified.

Author Response

POINT-BY-POINT RESPONSES TO REVIEWERS’ COMMENTS

POINT-BY-POINT RESPONSES TO REVIEWER #2

Comment-1: Up-to-date references should be used.

Responses – We indeed appreciate the suggestions. Accordingly, we have up-dated the references except for the conventional but necessary ones in the revised manuscript.

Comment-2: DOI numbers should be included whenever possible (line 186-187 of the writing guidelines).

Responses – We have provided the possible DOI numbers.

Comment-3: This paper makes important contributions to genetic diversity analysis and conservation strategies. However, issues such as genetic problems of modern varieties and climate change preparedness should be emphasized more.

Responses – Thank a lot for this comment. We have added a few more sentences to emphasize the challenges of modern varieties under climate change in Discussion of the revised manuscript. (Please see line 433-456).

Comment-4: Concrete suggestions for expanding the genetic base of modern varieties could be added.

Responses – Thank a lot for this comment. We have added some words to strongly propose the need of expanding the genetic base of modern varieties. (Please see line 577-583).

Comment-5: There could be broader discussions in the context of climate change.

Responses – We have added some more discussions to emphasize the importance of genetic resources for issues such as climate change. (Please see line 433-456).

Comment-6: The language could be slightly simplified.

Responses – We carefully edited the English language of the manuscript by the help of a native English speaker.

Reviewer 3 Report

Comments and Suggestions for Authors

The research is interesting and generates new information. I suggest following improvements:

1. Please indicate whether the SSR markers have been scored as codominant or dominant markers.

2. Use uniformity in describing genetic resources in Table 1. A 'variety' is different from a 'genetic stock' and 'germplasm'. Also indicate the parentage of the traditional and modern varieties.

3. It will be more useful if the results are discussed with the information of parentage. Some varieties might have common parent, which may be the basis of genetic relatedness. 

4. Marker informativeness is not reflected. The authors should dedicate a paragraph on marker information.

5. The sampling section should be dropped. The collection is not large enough to devise sampling strategy. 

Author Response

POINT-BY-POINT RESPONSES TO REVIEWERS’ COMMENTS

POINT-BY-POINT RESPONSES TO REVIEWER #3

General comment: The research is interesting and generates new information. I suggest following improvements:

Responses – We indeed appreciate the constructive comments from the Reviewer #3 and have modified our manuscript following these comments.

Comment-1: Please indicate whether the SSR markers have been scored as codominant or dominant markers.

Responses – Thank a lot for this comment. In fact, the SSR markers have been scored as codominant markers. We have added some explanations for this point in the revised manuscript. (Please see lines 156-158; lines 172-177).

Comment-2: Use uniformity in describing genetic resources in Table 1. A 'variety' is different from a 'genetic stock' and 'germplasm'. Also indicate the parentage of the traditional and modern varieties.

Responses – Thanks for the comment and we have modified the table caption in the revised manuscript. (Table 1).

Comment-3: It will be more useful if the results are discussed with the information of parentage. Some varieties might have common parent, which may be the basis of genetic relatedness.

Responses – Thanks for the suggestions. Since we only have limited number of varieties, we hesitated not expand our discussion too much in Discussion on the general parentage of all varieties. However, we only had limited discussions on the possible parentage of the modern varieties. (Please see lines 564-568).

Comment-4: Marker informativeness is not reflected. The authors should dedicate a paragraph on marker information.

Responses – Thanks a lot for this comment. We have added some relevant information for the SSR markers in the revised manuscript. (Please see lines 156-158; lines 172-177).

Comment-5: The sampling section should be dropped. The collection is not large enough to devise sampling strategy.

Responses – Thank for this comment. However, we feel that the proposal of a workable sample size to be used as a reference (not a rule) can be helpful for collectors working in the field. The proposed pattern of sampling numbers against their diversity changes of a naked barley variety is comparable with that of cultivated rice, both are inbreeding and relatively homozygous crops. Therefore, just for providing reference information, we would like to maintain this part in this manuscript for audience who might need such information.

Round 2

Reviewer 2 Report

Comments and Suggestions for Authors

acceptance of the article in this format is acceptable to me